Journal of Data-centric Machine Learning Research (2024)        Submitted 04/24; Revised 08/24; Published 08/24

# Forecasting Electric Vehicle Charging Station Occupancy: Smarter Mobility Data Challenge

**Yvenn Amara-Ouali**                                      YVENN.AMARA-OUALI@UNIVERSITE-PARIS-SACLAY.FR
*EDF R&D and Université Paris-Saclay*

**Yannig Goude**                                                           YANNIG.GOUDE@EDF.FR
*EDF R&D and Université Paris-Saclay*

**Nathan Doumèche**                            NATHAN.DOUMECHE@SORBONNE-UNIVERSITE.FR
*EDF R&D and Sorbonne Université*

**Pascal Veyret**                                                         PASCAL.VEYRET@EDF.FR
*EDF R&D*

**Alexis Thomas**                                       ALEXIS.THOMAS@MINES-PARISTECH.FR
*Ecole des Mines de Paris*

**Daniel Hebenstreit**                       DANIEL.HEBENSTREIT@STUDENT.TUGRAZ.AT
*Graz University of Technology*

**Thomas Wedenig**                             THOMAS.WEDENIG@STUDENT.TUGRAZ.AT
*Graz University of Technology*

**Arthur Satouf**                                            ARTHUR75FRANCE@GMAIL.COM
*CY Tech*

**Aymeric Jan**                                                            AJAN11@SLB.COM
*SLB, AI Lab*

**Yannick Deleuze**                                        YANNICK.DELEUZE@VEOLIA.COM
*Veolia S&TE*

**Paul Berhaut**                                          PAUL.BERHAUT@AIRLIQUIDE.COM
*Air Liquide*

**Sébastien Treguer**                                                STREGUER@GMAIL.COM
*INRIA*

**Reviewed on OpenReview:** *https://openreview.net/forum?id=0w0wrH5fXQ*

**Editor:** Holger Caesar

## Abstract

The transportation sector is a major contributor to greenhouse gas emissions in Europe. Shifting to electric vehicles (EVs) powered by a low-carbon energy mix could reduce carbon emissions. To support electric mobility, a better understanding of EV charging behaviours at different spatial and temporal resolutions is required, resulting in more accurate forecasting models. For instance, it would help users getting real-time parking recommendations, networks operators planning maintenance schedules, and investors deciding where to build new stations. In this context, the Smarter Mobility Data Challenge has focused on the development of forecasting models to predict EV charging station occupancy. This challenge

involved analysing a dataset of 91 charging stations across four geographical areas over seven months in 2020-2021. The forecasts were evaluated at three spatial levels (individual stations, areas regrouping stations by neighborhoods and the global level of all the stations in Paris), thus capturing the different spatial information relevant to the various use cases. The results uncover meaningful patterns in EV usage and highlight the potential of this dataset to accurately predict EV charging behaviors. This open dataset addresses many real-world challenges associated with time series, such as missing values, non-stationarity and spatio-temporal correlations. Access to the dataset, code and benchmarks are available at https://gitlab.com/smarter-mobility-data-challenge/tutorials to foster future research.

**Keywords:** Electric Mobility, Statistical Learning, Boosting, Hierarchical Forecasting

## 1 Introduction

**Electric mobility** The transportation sector is currently one of the main contributors to greenhouse gas emissions in Europe (IEA, 2022). To reduce these emissions, an interesting avenue has been to foster the development of EVs. In 2021, China led global EV sales with 3.3 million units, tripling its 2020 sales, followed by Europe with 2.3 million units, up from 1.4 million in 2020 (IEA, 2022). The U.S. market share of electric vehicles doubled to 4.5%, with 630,000 units sold. Meanwhile, electric vehicle sales in emerging markets more than doubled (IEA, 2022). As a consequence, electric mobility development entails new needs for energy providers and consumers (RTE, 2022). Companies and researchers are proposing a large amount of innovative solutions including pricing strategies and smart charging (Dallinger and Wietschel, 2012; Wang et al., 2016; Alizadeh et al., 2017; Moghaddam et al., 2018; Crozier et al., 2020) to couple it with renewable production Hafeez et al. (2023). However, their implementation requires a precise understanding of charging behaviours and better EV charging models are necessary to grasp the impact of EVs on the grid (Gopalakrishnan et al., 2016; Kaya et al., 2022; Ciociola et al., 2023; Andrenacci and Valentini, 2023). In particular, forecasting the occupancy of a charging station can be a critical need for utilities to optimise their production units according to charging demand (Zhang et al., 2023). On the user side, knowing when and where a charging station will be available is critical, but large-scale datasets on EVs are rare (Calearo et al., 2021; Amara-Ouali et al., 2021).

**Summary of the challenge** This article presents the Smarter Mobility Data Challenge, which aims at testing statistical and machine learning forecasting models to predict the states of a set of charging stations in the Paris area at different geographical resolutions. This challenge was held from October, 3rd 2022 to December 5th, 2022 on the CodaLab platform `https://codalab.lisn.upsaclay.fr/competitions/7192`. It was organised by the *Manifeste IA*, a network of 16 French industrials and TAILOR, a European project which aims to provide the scientific foundations for Trustworthy AI. It has been pioneered following the 'AI for Humanity' French government plan launched in 2019. The challenge gathered 169 participants and was open to students from the EU. The authors (except the participants of the challenge) have collected and prepared the dataset, and organised the data challenge.

**Time series models** Forecasting time series data is essential for businesses and governments to make informed decisions. However, the temporal structure in time series comes with specific challenges, such as non-stationarity and missing values. This is why, in addition to standard machine learning models, a wide range of models have been tailored for time series.

These include auto-regressive models (Box et al., 2015), tree-based models (Friedman, 2001), and deep learning models such as recurrent neural networks (Jordan, 1997; Hochreiter and Schmidhuber, 1997), temporal convolutional networks (Bai et al., 2018) and transformers (Wen et al., 2023). However, no one model has proven to be better than the others at predicting time series. On the one hand, although deep learning models are known to perform well with large datasets, it is still unclear how they compare to other models on small datasets, how they handle non-stationary data or how they deal with with exogenous information (Zeng et al., 2023; Kshitij et al., 2024). In fact, modern machine learning models still struggle to deal with missing values and time-dependent patterns such as trends or breaks. On the other hand, tree-based models such as gradient-boosted trees are known to perform well on tabular data (McElfresh et al., 2023), and to sometimes outperform complex deep learning models (Makridakis et al., 2022a). Therefore, practical insights from datasets and benchmarks are valuable (Petropoulos et al., 2022). In particular, a recent comprehensive benchmark (Godahewa et al., 2021) has regrouped 26 time series datasets on various domains, including energy and transport, taken from challenges (see, e.g., Makridakis et al., 2022b) and the public domain. Other works have proposed synthetic datasets to evaluate specific properties of forecast algorithms, such as interpretability (Ismail et al., 2020), outlier detection (Lai et al., 2021), and forecast performance (Kang et al., 2020).

**Hierarchical forecasting**   The data of the Smarter Mobility Data Challenge has a hierarchical structure because it EV charging stations can be regrouped at different scales (stations, areas, and global). Hierarchical time series forecasting has been studied on various other applications where the data is directly or indirectly hierarchically organised. For example, in the retail industry, goods are often classified into categories (such as food or clothing) and inventory management can be done at different geographical (national, regional, shop) or temporal (week, month, season) scales. Moreover, electricity systems often have an explicit (electricity network) or implicit (e.g., customer types, tariff options) hierarchy. Recent work shows that exploiting this structure can improve forecasting performance at different levels of hierarchy. For instance, Hyndman et al. (2011) focuses on tourism demand, Athanasopoulos et al. (2020) on macroeconomic forecasting, and Hong et al. (2019); Brégère and Huard (2022); Taieb et al. (2020); Nespoli and Medici (2022) on electricity consumption data.

**Related works**   Similar to energy and transport forecasting, EV demand forecasting has received a lot of attention. The survey by Amara-Ouali et al. (2022) compares the classical time series methods, the statistical models, the machine learning methods and the deep learning methods that have been used to capture the temporal dependencies in EV charging data. Overall, it shows that both tree-based models and deep learning models are able to capture the complex non-linear temporal relationships in EV charging data. More recently, Ma and Faye (2022) proposed a hybrid LSTM model that outperformed classical machine learning approaches (support vector machine, random forest, and Adaboost) and other deep learning architectures (LSTM, Bi-LSTM, and GRU) in forecasting the occupancy of 9 fast chargers in the city of Dundee. Wang et al. (2023) have investigated the use of spatial correlations to predict EV charging behaviour. They proposed a spatio-temporal graph convolutional network incorporating both geographical and temporal dependencies to predict the short-term charging demand in Beijing using a dataset of 76774 private EVs. However, such individual data is expensive and often kept private, and Wang et al. (2023) only had

access to data for the month of January 2018. In fact, although datasets describing the development of EV infrastructures are common (see, e.g., Falchetta and Noussan, 2021; Yi et al., 2022), fewer datasets document the actual use of EVs and they are often of lower spatial resolution (see, e.g., Lee et al., 2019). In fact, open datasets at the scale of individual stations, such as the one presented in this article, are still very rare (Amara-Ouali et al., 2021). Such so-called EVSE-centric (for Electric Vehicle Supply Equipment) datasets are more informative and hierarchical forecasting could be useful for users and operators interested in specific EV stations. However, even with EV datasets spanning multiple years, ruptures are common and models require specific adjustments (see, e.g., Koohfar et al., 2023).

**Main Contributions**    The main contributions of the paper can be summarised as follows:

1. An open dataset on electric vehicle behaviors gathering both spatial and hierarchical features, available at `https://gitlab.com/smarter-mobility-data-challenge/additional_materials`. Datasets with such features are rare and valuable for electric network management.
2. An in-depth descriptive analysis of this dataset revealing meaningful user behaviors (work behaviors, daily and weekly patterns...).
3. A detailed and reproducible benchmark for forecasting the EV charging station occupancy. This benchmark compares the winning solutions of a data challenge and state-of-the-art predictive models.

**Overview**    The paper is structured as follows. Section 2 describes the dataset. Section 3 details the forecasting problem at hand and baseline models. Section 4 presents the methods proposed by the three winning teams. Finally, Section 5 summarizes the findings and discusses our results. The full dataset, baseline models, winning solutions, and aggregations, are available at https://gitlab.com/smarter-mobility-data-challenge/tutorials and distributed under the Open Database License (ODbL). A supplementary material presents the Belib's pricing and park history in Section 1, a detailed data description (collection, preprocessing, explanatory data analysis) in Section 2, some complements about the winning strategies of the challenge in Section 3, future perspectives about new datasets and benchmarks in Section 4 and a Datasheet in Section 5.

## 2 EV charging dataset

In this section we present how the raw dataset was collected and how it was then preprocessed to make it suitable for the data challenge.

**General description**    The dataset is based on the real-time charging station occupancy information of the Belib network, available on the Paris Data platform (ODbL) (of Paris, 2023). The Belib network was composed of 91 charging stations in Paris at the time of the challenge, each offering 3 plugs for a total of 273 charging points. A process to store the data was initiated by the EDF R&D team since daily data was not stored by Paris Data. A pipeline was set up to collect this data every 15 minutes, starting July 2020, on the platform's dedicated API `https://parisdata.opendatasoft.com/explore/dataset/belib-points-de-recharge-pour-vehicules-electriques-disponibilite-temps-reel/api`. The data was then stored in a data lake based on Hadoop technologies (HDFS, PySpark, Hive,

and Zeppelin). The storage of this information over time allows, for example, to estimate the usage of the charging stations depending on their location.

**Belib's history: pricing mechanism and park evolution**  89% of EV users living in a house mainly charge their vehicle at home, compared to only 54% of EV users living in an apartment in 2020 (ENEDIS, 2021). Paris is a very dense city that allows limited access to private residential charging points, hence the need for public charging stations. The first 5 stations of the Belib network were commissioned on 12 January, 2016 (Torregrossa, 2016; Camille, 2016). The network grew progressively in 2016 to reach 60 stations all around Paris. Users needed to buy a 15 euro badge to connect to the network. Different pricing strategies were applied depending on the time of the day and plugs. The "normal charge" of 3kW was free at night (between 8 p.m. and 8 a.m.) and cost 1 euro per hour on daytime (between 8 a.m. and 8 p.m.). The "quick charge" of 22kW cost 25 cents every 15 minutes during the first hour of charge. After the first hour, the first 15 minutes cost 2 euros. After this 1h and 15 minutes, each 15 minutes cost 4 euros. Each station contained 3 parking spots:

- one dedicated to "normal charge" with an E/F electric plug,

- one dedicated to "quick charge" with a ChaDeMo and a Combo2 plugs,

- one where both "normal charge" and "quick charge" were possible, with an E/F, a T2, and a T3 plugs.

The pricing strategy was intended to allow the usage of "normal charge" plugs as a free parking spot overnight, while "quick charge" became expensive after one hour of usage.

In 2021, the city of Paris allowed the company TotalEnergies to run the Belib network for a period of 10 years. The goal is to enhance the network, increasing from 90 stations and 270 charging points, to 2300 charging points (TotalEnergies, 03/31/2021; Livois, 04/09/2021). We elaborate on the new pricing mechanism in the supplementary material.

**Data preprocessing**  In the raw data, each observation reflects the status of the plugs (up to 6) within a charging point. The structure of the raw dataset is misleading as only one of these plugs can be in use at a time. Therefore, we processed the dataset to only keep the relevant rows, i.e., the rows containing the plugs in use, and we treated a charging point as a single plug. In addition, charging points are clustered in groups of three according to their geographic location in the raw data. This charging point structure was confirmed by the data provider. We grouped the three adjacent charging points into a single charging station and aggregated the data accordingly. To account for differences in timestamp synchronization between stations, we have adjusted timestamps to match the nearest 15-minute interval. The *available*, *charging*, and *passive* states are taken directly from the raw data. The last state *other* regroups several statuses including *reserved* (a user has booked the charging point), *offline* (the charging point is not able to send information to the server), and *out of order* (the charging point is out of order). We made this choice because of the relatively small number of *reserved* and *out of order* records. This way, the *other* state could be interpreted as a noisy version of the *offline* state. Missing timestamps in the dataset have not been filled so there is room for missing data imputation techniques.

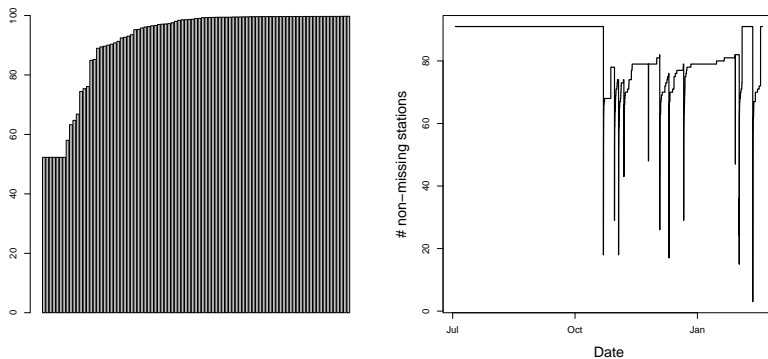

Figure 1: Left: Percentage of non missing observations per station. Right: Number of non missing stations in function of time on the train set.

**Missing values** There is a significant number of missing values in the data. To illustrate, the records of the following five days are very incomplete as they contain less than 96 observations for all the 91 stations: 2020-08-06 (with 92 data points), 2020-10-27 (95), 2020-11-20 (54), 2020-12-29 (95), and 2021-01-04 (95). The distribution of missing values is highly station-dependent, as illustrated in Figure 1. We note that half of the stations have almost no missing data (except for the five days documented above), whereas 7 stations have around 50% missing observations. This suggests that malfunctioning behaviors are specific to some stations and could be learned. In addition, the number of non-missing stations is depicted in Figure 1. Note that we excluded timestamps from the plot when all stations were missing. We also note that the number of missing stations starts to fluctuate a lot after October.

**Exploratory Data Analysis** We show daily and weekly profiles with the median number of plugs as a function of time (an instant corresponding to a 15-minute interval) per status at the Global level on Figure 2. From these graphs, We observe the presence of a daily pattern in the data and a change in the pattern between weekdays and weekends. What we observe in Figure 2 matches with the pricing strategy used from 2016 to 2021, detailed in Paragraph 2. The pricing changed twice a day: at 8 a.m. and 8 p.m. At night, the free "normal charge regime" (7kW) explains the peak in *charging* states at instant 80 (corresponding to 8 p.m.) and the drop of *available* at the same hour. This "normal charge" mode provides a low electrical power, hence the slowness of charging. Therefore, overnight, as EV batteries become fully charged, the number of *charging* states decrease in favour of the number of *passive* states. The proportion of *other* states is more important at night, mainly because there is more maintenance jobs at night. Since users tend to be repelled by or ignore malfunctioning stations it explains why this excess in *other* comes with a slight increase of *available* at night. On the other hand, the price increase, after 8 a.m., induces a decrease of *passive* spots from 7p.m. to 9 p.m. and an increase of *available* (drivers parking on regular parking spots) and *charging* spots (drivers charging their car in front of their office after the morning ride). This analysis is consistent with the weekly scale in Figure 2. The number of *charging* stations is greater during work days, while the number of *available* stations is greater during week-ends,

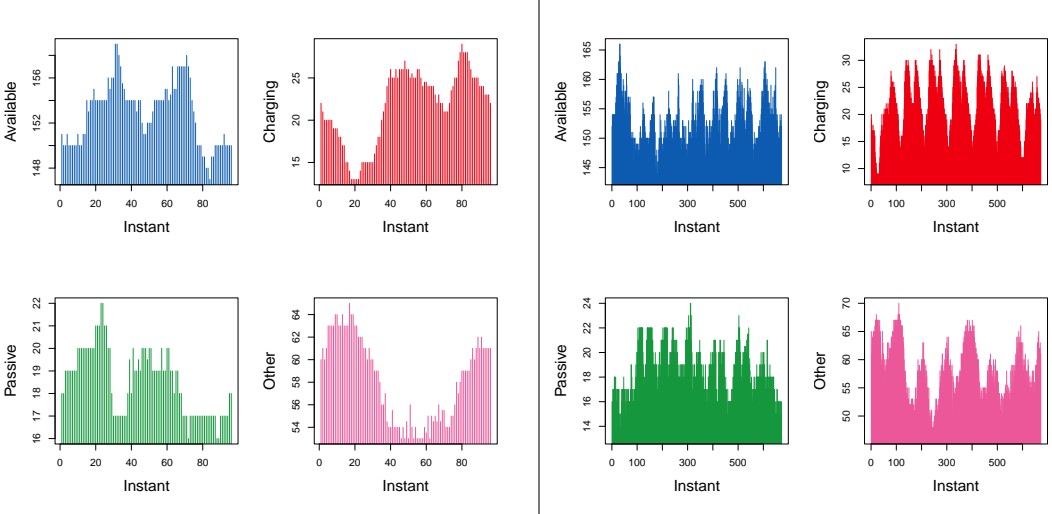

Figure 2: Daily (left) and Weekly (right) profiles for each status at the Global level.

reflecting commuting behaviors. We note that the daily peaks at 8 a.m. and 8 p.m. are pronounced on the weekly *charging* profile.

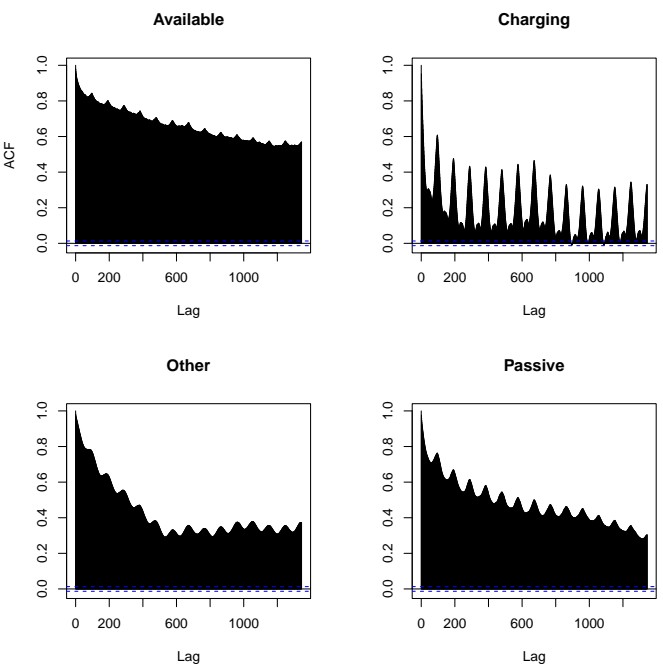

Figure 3: Empirical ACF of the 4 status at the global level.

Figure 3 shows the empirical autocorrelation functions (ACF) at the global level. As excepted, we observe daily and weekly cycles. The daily cycle depends on the state of the plug. The non-stationnarity of the data is visible on the ACF: the *available* status slowly decreases

on its ACF due to the low frequency component of the data. We study the distribution of the states with respect to time and stations. The barplots of the corresponding frequencies (in percent) are shown in Figure 4 (left). We note a major difference between the *available* status and the 3 others; the stations' plugs are most often available than in any other state. The distribution of the 4 states by area is shown in Figure 4 (right). The distribution profile is similar in all areas with a high frequency of *available* status, followed by *other*, *passive* then *charging*. We note that the *other* status is over represented in the north area. The west area has lower availability due to higher charging activities as well as a high representation of Other. The south and east area are very similar, with higher representation of the *available* status.

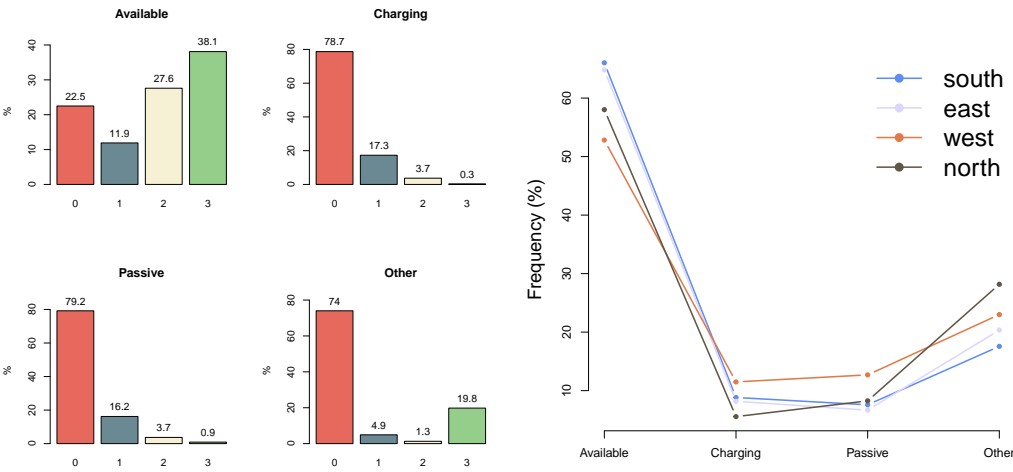

Figure 4: Left: distribution of the 4 states over all the stations and instants. Right: distribution of the 4 states by area.

## 3 Problem description

In this section, we introduce the hierarchical forecasting challenge proposed to the contestants of the Smarter Mobility Challenge. The overall goal is to forecast the occupancy of charging stations at different geographical resolutions: single station, regional and global Paris area. Accurate prediction of a single station typically benefits to EV drivers looking for available charging points, whereas forecasting the occupancy of a network of charging stations allows utility providers to optimise their production units. This can lead to significant savings for the electricity system (around 1 billion euros per year, see RTE (Sections 5.4 and 5.5, 2019) and Lauvergne et al. (2022)).

**Data splitting**   For this data challenge, we split the data between a training and a testing set. Because of the change of operator and pricing (see Section 2) on March 25th, 2021, we decided to study the following period: from July 3rd, 2020 to March 10th, 2021, when both the EV park and the pricing stayed unchanged. To mimic a genuine time-series forecasting problem, we preserved the time structure when partitioning the data and selected a test set

of three weeks. The test set is a stable period that does not include significant changes in the data on the global level (Figure 5).

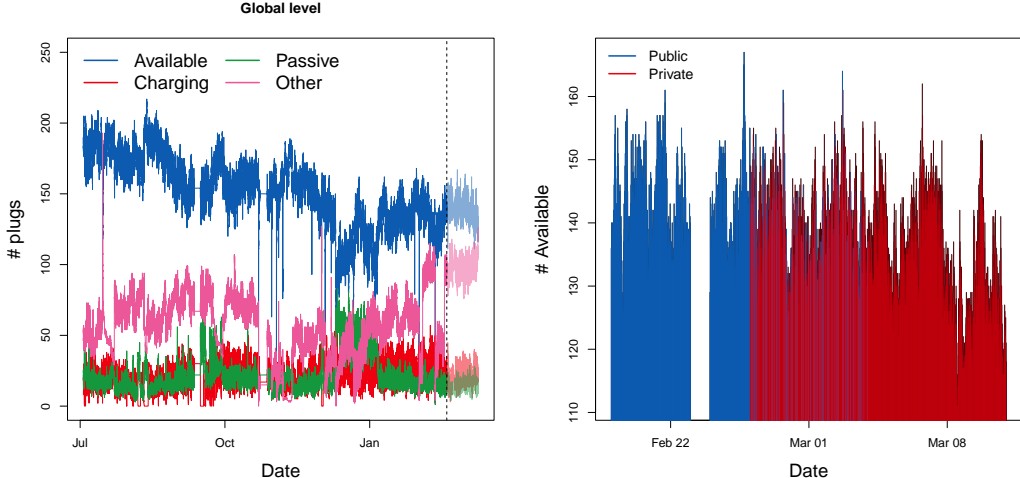

Figure 5: Left: total number of plugs in each state in function of time on train and test (transparent color). The vertical dashed lines represents the end of the train set. Right: total of *available* plugs in function of time on the test set. In blue: public set. In red: private set.

The training set contains $D_{train}$ points from 2020-07-03 00:00 to 2021-02-18 23:45. The test set contains $D_{test}$ points from 2021-02-19 00:00 to 2021-03-10 23:45. As most EVSE stakeholders (e.g., EDF Group) receive the data with a delay of one to two weeks, we designed the challenge to match the operational perspective, hence the two-week forecast horizon. The test set has been divided into two subsets: a public set for validation purposes and a private set $D_{private}$. The latter being used to quantify the performance of the solution while minimising the risk of overfitting.

To create the public and the private sets, the test set was split into three subsets of one week each. The first week was assigned to the public set, and the third one to the private set. We randomly assigned 20% of second week to the public set and the rest to the private, as illustrated in Figure 5. February 23 was excluded of the test set as it contains outliers. The public and private test sets were structured to balance the preservation of the temporal structure of the data and to avoid overfitting on short forecast horizons.

**Target description**  At any given time, a plug is in one of the four states.

- A station is in state $c$ (*charging*) when it is plugged into a car and provides electricity.

- In state $p$ (*passive*) when connected to a car that is already fully charged.

- In state $a$ (*available*) when the plug is free.

- In state $o$ (*other*) when the plug is malfunctioning.

We denote by $y_{t,k} = (a_{t,k}, c_{t,k}, p_{t,k}, o_{t,k}) \in \{0, 1, 2, 3\}^4$ the vector representing the state of station $k \in \{1, \ldots, 91\}$ at time $t$, where $a_{t,k}$ is the number of available plugs, $c_{t,k}$ the number

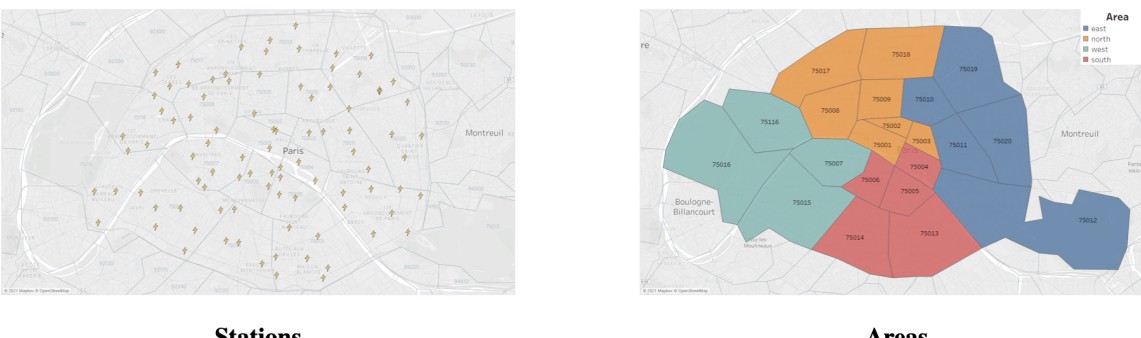

**Stations**                    **Areas**

Figure 6: The 91 stations (yellow dots on the left) and the 4 areas of Paris (colored on the right)

of charging plugs, $p_{t,k}$ the number of passive plugs, and $o_{t,k}$ the number of other plugs, at station $k$ and time $t$. By definition, eq. 1 is always valid,

$$a_{t,k} + c_{t,k} + p_{t,k} + o_{t,k} = 3. \tag{1}$$

**Features description**   To predict the state of station $k$ at time $t$, the dataset contains the following variables:

- Temporal information: *date*, *tod* (time of day), *dow* (day of week), and *trend* (a temporal index).

- Spatial information: *latitude*, *longitude*, and *area* (south, north, east, and west) of the station.

*dow* is the day of week (from 1 for Monday to 7 for Sunday) and *tod* the time of day, by interval of 15 minutes (0 for 00:00:00 to 95 for 23:45:00). The *trend* feature is the numerical conversion of the time index, and *date* is the corresponding string, in the ISO 8601 format. The data is then aggregated into 4 areas of about 20 stations each, as shown in Figure 6.

**Evaluation**   We aim to forecast the state of the different plugs at 3 hierarchical levels:

- Individual stations: denoted by $y_{t,i}$, for $i \in \{1 \dots 91\}$.

- Areas, corresponding to the cardinal points: $y_{t,\text{south}}$, $y_{t,\text{north}}$, $y_{t,\text{east}}$, and $y_{t,\text{west}}$

- At the global level: $y_{t,\text{global}}$

we also introduce $y_{t,\text{zone}} = \sum_{i \in \text{zone}} y_{t,i}$ as the sum of the plugs per state in a zone (south, north, east, west, or global). Let $z_t = (y_{t,1}, \dots, y_{t,91}, y_{t\text{south}}, y_{t,\text{north}}, y_{t,\text{east}}, y_{t,\text{west}}, y_{t,\text{global}})$ be the aggregated matrix containing the statutes of all stations at the different hierarchical levels at time $t$. The goal is to provide the best estimator $\hat{z}$ of $z$. Performance is evaluated using the following score, encoding each hierarchical level as a penalty.

$$L(z, \hat{z}) = |D_{\text{private}}|^{-1} \sum_{t \in D_{\text{private}}} \left( \ell_{\text{station}}(z_t, \hat{z}_t) + \ell_{\text{area}}(z_t, \hat{z}_t) + \ell_{\text{global}}(z_t, \hat{z}_t) \right), \tag{2}$$

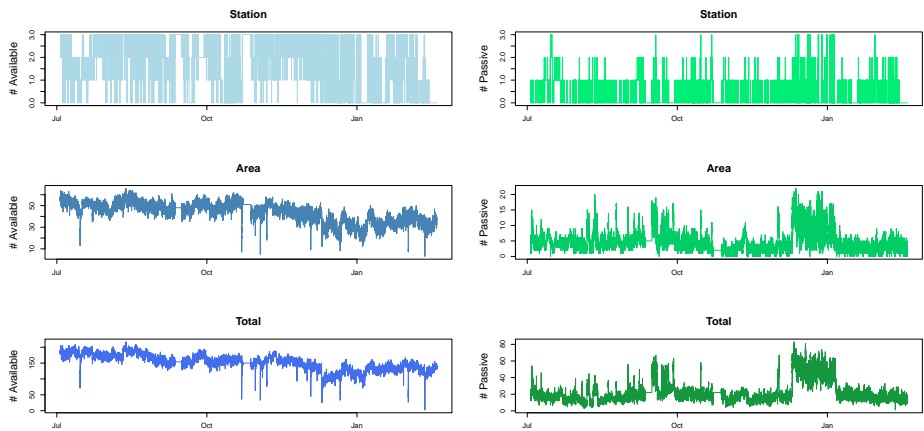

Figure 7: Number of available (left) an passive (right) plugs in function of time for one station, its corresponding area and at the global level.

with the different terms defined as follows:

$$\ell_{\text{station}}(z_t, \hat{z}_t) = \sum_{k=1}^{91} \|y_{t,k} - \hat{y}_{t,k}\|_1,$$

$$\ell_{\text{area}}(z_t, \hat{z}_t) = \sum_{\text{zone} \in \mathcal{C}} \|y_{t,\text{zone}} - \hat{y}_{t,\text{zone}}\|_1,$$

$$\ell_{\text{global}}(z_t, \hat{z}_t) = \|y_{t,\text{global}} - \hat{y}_{t,\text{global}}\|_1,$$

where $\mathcal{C} = \{\text{south}, \text{north}, \text{east}, \text{west}\}$ is the set of cardinal points and $\|x\|_1 = \sum_{k=1}^{p} |x_k|$ is the usual $\ell^1$ norm on $\mathbb{R}^p$. We illustrate the different hierarchical level of the data in Figure 7. We observe that spatial aggregation increases the signal-to-noise ratio, as the variance tends to decrease when the spatial aggregation is broader.

**Baseline models**   As a baseline, we provided two models. A first naive estimator of $z_t$ is the median per day of week and quarter-hour over the training set, in which we removed the missing values:

$$\hat{z}_t = \underset{t' \in Cal_t}{\text{median}}\{z_{t'}\}, \tag{3}$$

where

$$Cal_t = \{t' \in D_{train}, \ \text{dow}(t') = \text{dow}(t)\} \cap \{t' \in D_{train}, \ \text{tod}(t') = \text{tod}(t)\}.$$

Notice that the $Cal_t$ corresponds to the timestamps of the same day of the week and the same hour of the day.

The second baseline model is the parametric model called (CatBoost). It is a tree-based gradient boosting algorithm designed to solve regression problems on categorical data. We used its implementation in the python library `CatBoost` (Prokhorenkova et al., 2018) and it has demonstrated excellent performance for a great variety of regression tasks (Daoud, 2019; Huang et al., 2019; Hancock and Khoshgoftaar, 2020) and forecasting challenges (Makridakis et al., 2022b). The performance of these two baselines on the private test set is shown by the dotted lines in Figure 8, next to the solutions of the winning team.

## 4 Solutions of the winning teams

This section describes the methods used by the three winning teams. The ranking of the top competitors is shown in Figure 8. The confidence intervals are constructed by time series bootstrapping (non-overlapping moving block bootstrap) (Kunsch, 1989; Politis and Romano, 1994). One subsection is dedicated to each of the winning teams, as their approaches are informative for the analysis of the dataset. In the last subsection, their strengths are combined using aggregation methods.

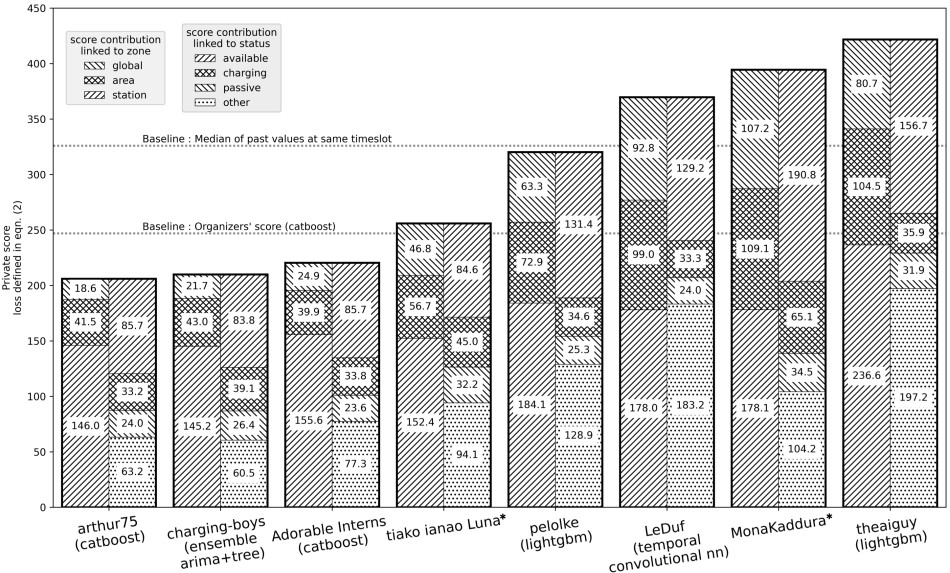

Figure 8: Ranking of the top competitors.
* No information about these methods were provided by these competitors.

### 4.1 Arthur Satouf (team Arthur75)

**Data exploration** As shown in Figure 1, the dataset presents a lot of missing data. Common techniques were considered to impute these Pratama et al. (2016), including computing the mean by station, forward and backward filling, simple moving average, weighted moving average, and exponential moving weighted average (EMW) Hunter (1986). These techniques are evaluated by measuring the mean absolute error (MAE) on a validation subset of the training set. As a result, the EMW is the most effective technique, and it is thus implemented for both forward and backward filling approaches. Specifically, we use the last 8 known values to forward fill the first 8 missing values. The same procedure is applied to backward filling.

Table 1: Example of a data conversion to a string

| Given station at a given time | Available | Charging | Passive | Other | Target |
|:---:|:---:|:---:|:---:|:---:|:---:|
| 14h15-16/08/2021 | 1 | 2 | 0 | 0 | 1200 |
| 14h30-16/08/2021 | 0 | 1 | 1 | 1 | 0111 |
| 14h45-16/08/2021 | 0 | 0 | 3 | 0 | 0030 |

**Model description**    We compare usual forecasting models Ahmed et al. (2010); Chen and Guestrin (2016); Ribeiro and dos Santos Coelho (2020), such as SARIMAX, LSTM, XGBoost, random forest, and CatBoost. The evaluation metric used is the MAE, and the time series cross-validation technique is applied to evaluate the performance of the models Kreiss and Paparoditis (2011); Pedregosa et al. (2011). The CatBoost algorithm is ultimately chosen for its fast optimization relying on parallelization and its ability to handle categorical data without preprocessing. As explained in Section 2, the states of any station $k$ satisfy at any time $t$ the equation $a_{t,k} + c_{t,k} + p_{t,k} + o_{t,k} = 3$, which is enforced in the CatBoost estimator as follows.

- At the station level, the problem is transformed from a multi-task regression problem to a classification problem. This is achieved by concatenating the values of each task as a string, resulting in 20 unique classes. In this approach, the sum of the four vectors always equals three, given that there are three plugs. After predicting a given target, the target is decomposed into four values. Table 1 provides an example.

- At the area level, CatBoost was also used as a regression problem, as shown in Figure 9 and Figure 10. However, each area had its own model, and each area used a combination of CatBoost regressor and Regressor-Chain Read et al. (2009). Regressor-Chain involves building a unique model for each task and using the result of each task as an input for the next prediction model. The output of each model, along with the previous output, is then used as input for the next task. This approach helps to keep the sum of plug equal to the right number and takes into account the correlation between tasks, making the prediction more robust.

- At the global level, the approach is similar to the one applied to the area level, with only 4 models as there are no longer areas.

A time series cross validation is used once again to tune the hyperparameters and to validate the models. It relies on the mean absolute percentage error Myttenaere et al. (2016) at the area and the global levels, and on the F-measure Chen et al. (2004) at the station level. In total, 21 CatBoost models are used to forecast the private datasets.

## 4.2  Thomas Wedenig and Daniel Hebenstreit (team Charging-Boys)

**Data exploration**    Exploratory experiments did not show any signs of a trend within the time series. Regarding stationarity, we run the Augmented Dickey–Fuller test Dickey and Fuller (1979) on the daily averages of the target values for each station and find inconclusive results. Therefore, we cannot assume stationarity for all target-station pairs, which is why

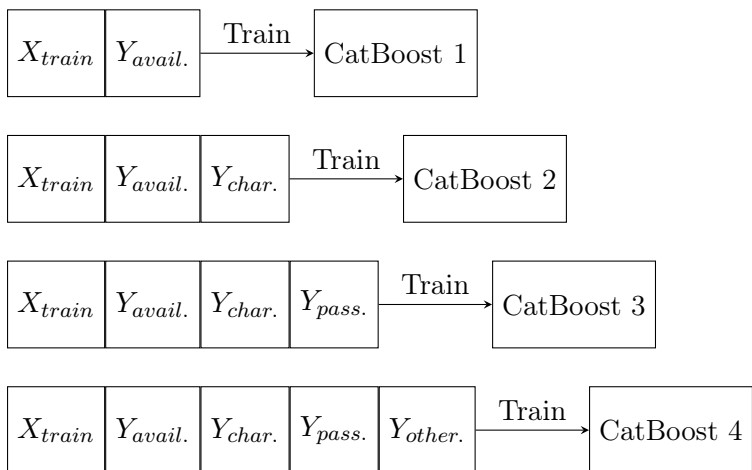

Figure 9: Training process of the regressor Chain with CatBoost-Regressor.

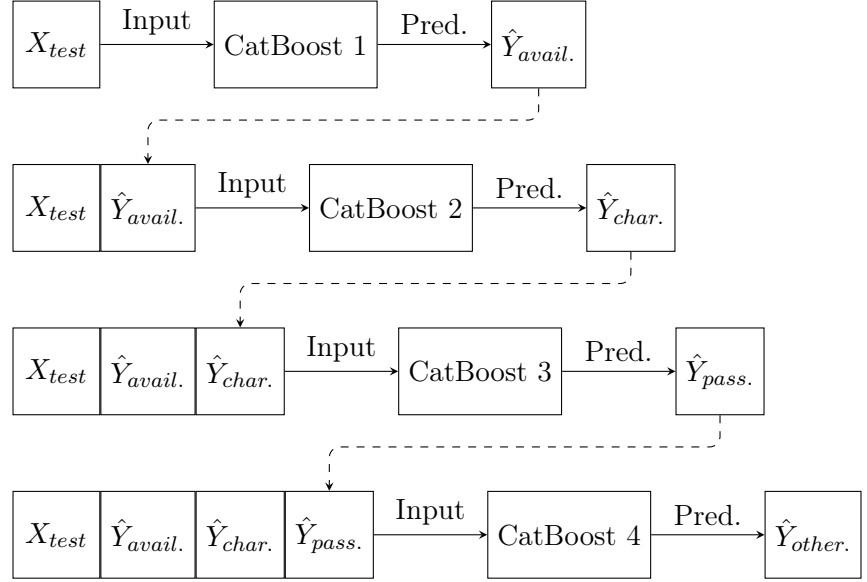

Figure 10: Inference process of the regressor Chain with CatBoost-Regressor.

we employ differencing in the construction of our ARIMA model. As usual in statistical frameworks, we assume that the noise interferes with the high frequencies of the signal. To denoise, we preprocess the time series by computing a rolling window average with a window size of 2.5 hours Hyndman and Athanasopoulos (2018). During our data exploration, we encounter a significant change in the behavior of the individual stations in the end of October 2020, just before the COVID-19 regulations were enforced in Paris. We also assume that several stations were turned off after this event, as labels were missing over large time intervals. Thus, we experiment with different methods of missing value imputation, but find that simply dropping the timestamps with missing values performs best. We add custom features, namely a column indicating whether the current date is a French holiday, as well as

sine and cosine transforms of *tod*, *dow*, the month, and the position of the day in the year. To ensure that our regression models return integer outputs that sum to 3 for each station and timestamp (since stations have exactly 3 plugs), we round and rescale these predictions in a post-processing step.

**Model description**   We train different models and then aggregate them. First, we start by considering a tree-based regression model. Using `skforecast` Rodrigo and Ortiz (2023), we train an autoregressive XGBoost model Chen and Guestrin (2016) with 100 estimators. We train it on all of the 91 stations individually, each having 4 targets, resulting in 364 models. Each model receives the last 20 target values, as well as the sine/cosine transformed time information as input, and predicts the next target value. We also discard all features that are constant per station (e.g., station name, longitude, and latitude). The final regression model achieves a public leaderboard score of 177.67.

Then, we consider a tree-based classification model. To effectively enforce structure in the predictions (i.e., that they sum to 3), we transform the regression problem discussed above into a classification problem. For a given station and timestamp, consider the set of possible target values $\mathcal{C} = \left\{ \mathbf{x} \in \{0, 1, 2, 3\}^4 \quad \text{s.t.} \ \sum_{i=1}^{4} x_i = 3 \right\}$. We treat each element $c \in \mathcal{C}$ as a separate class and only predict class indices $\in \mathcal{I} = \{0, \ldots, 19\}$ (since $|\mathcal{C}| = 20$). While $\mathcal{I}$ loses the ordinal information present in $\mathcal{C}$, this approach empirically shows competitive performance. When training a single XGBoost classifier with 300 estimators for all stations, we achieve a public leaderboard score of 178.9. We also experiment with autoregressive classification (i.e.,including predictions of previous timestamps), but find no improvement in the validation error.

Finally, we fit a non-seasonal autoregressive integrated moving average (ARIMA) model Box et al. (2015) for each target-station combination.

To predict the value of a given target, we only consider the last $p = 2$ past values of the same target (in the preprocessed time series) and do not use any exogenous variables for prediction (e.g., time information). We apply first-order differencing to the time series ($d = 1$) and design the moving average part of the model to be of first-order ($q = 1$). On the validation and training sets, forecasts were applied recursively, using past forecasts as ground truth.

We observe that the forecasts using these models have very low variance, i.e., each model outputs an approximately constant time series. These predictions achieve a competitive score on the public leaderboard (third place).

The final model is an ensemble of the tree-based regression model, the tree-based classification model, and the ARIMA model. For a single target, we compute the weighted average of the individual model predictions (per timestamp). The ensemble weights are chosen to be roughly proportional to the public leaderboard score ($w_{\mathrm{reg}} = 0.35$, $w_{\mathrm{class}} = 0.25$, $w_{\mathrm{ARIMA}} = 0.4$). Since the predictions of the tree-based models have high variance, we can interpret mixing in the ARIMA model's predictions as a regularizer, which decreases the variance of the final model. As the tree-based models also use time information for their predictions, we use the entirety of the available features.

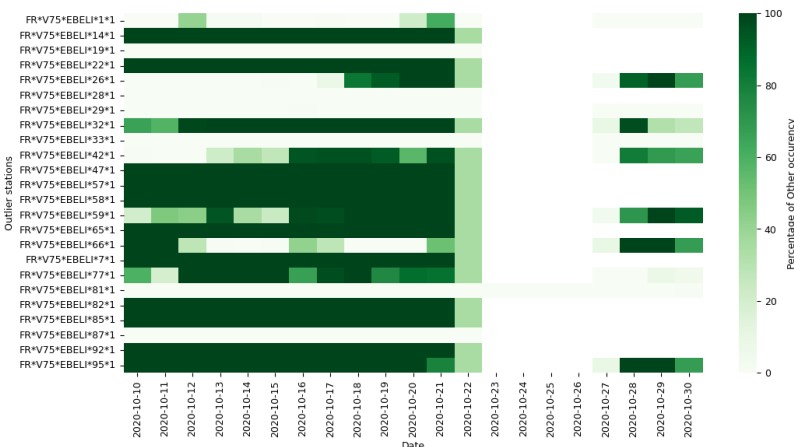

Figure 11: Percentage of state $o$ occurrences per outlier per day around 2020-10-22

### 4.3 Nathan Doumèche and Alexis Thomas (team Adorable Interns)

**Data exploration**    Several challenges arise from the data, as shown in Figure 1. An interesting phenomenon is the emergence of a change in the data distribution on 2020-10-22, characterized by the appearance of missing data. A reasonable explanation is that the detection of missing values is due to an update in the software that communicates with the stations. The update would have taken place on 2020-10-22, allowing the software to detect new situations in which stations were malfunctioning. This hypothesis is supported by the fact that the stations with missing values are those that were stuck in states corresponding to the absence of a car, i.e., either the state $a$ or the state $o$ (see Figure 11). In fact, 88% of the stations that were stuck in either $a$ or $o$ for the entire week before 2020-10-22 had missing values on 2020-10-22. Perhaps the users avoided the malfunctioning stations, or perhaps the users tried to connect to the station, but the plug was unresponsive, so the users went undetected. An important implication of this hypothesis is that the data before the change should not be invalidated, since the behaviour of the well-functioning stations did not change. Another challenge of the dataset was its shortness. In fact, we expect a yearly seasonal effect due to holidays (Xing et al., 2019) that cannot be distinguished from a potential trend because there is less than one year of data. All these observations suggest giving more weight to the most recent data.

As usual in the supervised learning setting, we need to choose a model $\mathcal{F}$ to construct the estimator $\hat{z}_t \in \mathcal{F}$. To estimate the entire $D_{test}$ period at once, we cannot rely on online models such as autoregressive models or hidden-state neural networks (RNN, LSTM, transformers...), although they perform well for time series forecasting (Bryan and Stefan, 2021), and in particular for EV charging station occupancy forecasts Ma and Faye (2022); Mohammad et al. (2023).

Once a model $\mathcal{F}$ is chosen, we define an empirical loss $L$ on the training data. Then, a learning procedure, such as a gradient descent, fits the estimator $\hat{z}$ that minimizes $L$, with the hope that $\hat{z}$ will minimize the expectation of the test loss (2) (Vapnik, 1991; Hastie et al., 2017). Given a training set $T_{train} \subseteq D_{train}$, we consider two empirical losses.

The first one corresponds to Eq. 4, this loss gives equal weight to all data points.

Table 2: Evaluation of the performance of the Adorable Interns' models in both phases

|  | Mean | Median | $C(4, 150)$ | $C_{exp}(5, 200)$ |
| --- | --- | --- | --- | --- |
| Benchmark Phase | 316 | 309 | 292 | **261** |
| Validation Phase | 323 | 303 | 233 | **189** |

$$L_{equal}(\hat{z}) = |T_{train}|^{-1} \sum_{t \in T_{train}} \|z_t - \hat{z}_t\|_1 \qquad (4)$$

The second one is given in Eq. 5.

$$L_{exp}(\hat{z}) = \sum_{t \in T_{train}} \exp((t - t_{max})/\tau)\|z_t - \hat{z}_t\|_1, \qquad (5)$$

where $\tau = 30$ days and $t_{max} = $ 2021-02-19 00:00:00.

This time-adjusted loss function is common for non-stationary processes Ditzler et al. (2015) because it gives more weight to the most recent observations. This makes it possible to give more credit to the data after the change in the data distribution and to capture the latest effect of the trend, while using as much data as possible.

**Model description** To compare the performance of the models, we defined a training period $T_{train}$, covering the first 95% of $D_{train}$, and a validation period $T_{val}$, covering the last 5%. In this benchmark phase, models are trained on $T_{train}$ to minimize $L_{equal}$ or $L_{exp}$, and then their performance is evaluated on $T_{val}$ by $L_{val}(\hat{z}) = |T_{val}|^{-1} \sum_{t \in T_{val}} \|z_t - \hat{z}_t\|_1$.

The $Mean$ model estimates $\hat{y}_{t,k}$, $\hat{A}_{t,k}$ and $\hat{G}_t$ by their mean over the training period for each value of $(tod, dow)$. Idem for the $Median$ model. They are robust to missing values since the malfunctioning of a station $k$ only affects $\hat{y}_{t,k}$.

We compare them with the CatBoost model presented in Section 3. Let $C(d, i)$ be the CatBoost model of depth $d$ trained with $i$ iterations using $L_{equal}$, and $C_{exp}(d, i)$ the same model trained using $L_{exp}$. In this setting, we train twelve CatBoost models: one for each pair of state $(a, c, p, o)$ and hierarchical level.

After hyperparameter tuning, we found $C(4, 150)$ and $C_{exp}(5, 200)$ to be the best models in terms of tradeoff between performance and number of parameters, knowing that early stopping and a small number of parameters prevent overfitting (see, e.g., Ying, 2019). All of these models take advantage of the fact that malfunctioning stations tend to stay in specific states.

The contest organizers allowed participants to test their models on a subset $T_{val}$ of $D_{test}$. In this validation phase, we trained our best models on the entire $D_{train}$ period and tested them with the test loss (2). Table 2 shows that the ranking of the models is preserved. The submitted model was therefore $C_{exp}(5, 200)$. Note that this model is also interesting because its small number of parameters ensures robustness and scalability. In addition, tree-based models are quite interpretable, which is paramount for operational use Jabeur et al. (2021).

### 4.4 Aggregation of forecasts from the winning teams

Naive aggregations of uncorrelated estimators are known to have good asymptotic (Tsybakov, 2003) and online (Cesa-Bianchi and Lugosi, 2006) properties. In practice, they often achieve better performance than the individual estimators (see, e.g., Bojer and Meldgaard, 2021; McAndrew et al., 2021).

Table 3 shows the performance of the top 3 teams compared with two aggregation techniques. The *Total* score is the result of Equation (2), while the other scores are straightforward subdivisions of the loss by hierarchical level and by state. Standard deviations are estimated by moving block bootstrap. The uniform aggregation –denoted by *Uniform agg.*– corresponds to the mean of each team's prediction, while the weighted aggregation –denoted by *Weighted agg.*– is computed by gradient descent using the MLpol algorithm (Gaillard et al., 2014) to minimise the error on the training set. Notice how the weighted aggregation outperforms the other forecasts for the total loss, as well as for all the subdivisions of the loss. Note that the weighted aggregation of the 3 teams forecasts performs better than the weighted aggregation of any subsets of it (Arthur75+Charging Boys: 199, Arthur75+Adorable Interns: 203, Charging Boys+Adorable Interns:200). From these results, each team brings a significant contribution to the final score.

Table 3: Score by target of the top 3 teams and aggregations.

|  | Available | Charging | Passive | Other | Stations | Area | Global | Total |
|---|---|---|---|---|---|---|---|---|
| Arthur75 | 85.7 (2.7) | 33.1 (0.7) | 24 (0.6) | 63.3 (2.8) | 145.6 (1.4) | 41.8 (2.5) | 18.7 (4.8) | 206.1 (5.7) |
| Charging Boys | 83.9 (3.3) | 38.9 (0.6) | 26.3 (0.4) | 60.7 (3.4) | 145.3 (1.8) | 42.9 (3) | 21.7 (5.7) | 209.9 (6.8) |
| Adorable Interns | 85.7 (2) | 33.8 (0.7) | 23.6 (0.6) | 77.4 (2.7) | 155.4 (1.5) | 40.1 (2.8) | 25 (3.8) | 220.5 (5.1) |
| Uniform Aggregation | 82.9 (2.5) | 33.1 (0.7) | 22.1 (0.5) | 63.4 (2.7) | 141.1 (1.4) | 40.5 (2.9) | 20 (4.4) | 201.5 (5.4) |
| Weighted Aggregation | 82.3 (2.7) | 33 (0.7) | 22.4 (0.5) | 58.5 (2.9) | 137.1 (1.4) | 40.3 (2.9) | 18.7 (4.4) | 196.2 (5.4) |

### 4.5 Neural networks

Although participants proposed a wide variety of models, they mainly focused on classical time series models like ARIMA (see, e.g., charging-boys) and tree-based models (see, e.g., arthur75). Indeed, the only neural network proposed in the challenge was LeDuf's temporal convolutional neural network, inspired by Bai et al. (2018), and it performed poorly (see Figure 8). Therefore, in order to get a better overview of their potential strengths, we completed our benchmark with neural networks after the challenge. The code to reproduce these experiments is available at `https://gitlab.com/smarter-mobility-data-challenge/tutorials/-/tree/master/2.%20Model%20Benchmark`.

Indeed, Fully Connected Neural Networks (FCNNs) are known to be able to forecast EV demand (Boulakhbar et al., 2022; Ahmadian et al., 2023). The FCNN model we implemented predicts the status of individual stations. The forecasts for the area and the global levels are then derived in a bottom-up manner by summing the forecasts of the individual stations. In contrast to the CatBoost models, this bottom-up approach performed better than training a FCNN for each hierarchical level (station, area and global). The hyperparameters of the FCNN were then optimised using the `optuna` package in `Python` (Akiba et al., 2019). As a result, the package selected a FCNN with one hidden layer, 155 neurons, a learning rate of $7.8e-4$, a dropout of 0.012, a batch size of 480 and 14 epochs. Similar to Ahmadian et al.

(2023), we found out that FCNNs with a single hidden layer were the ones that performed best. The performance of the FCNN on the test set for the hierarchical loss is $250.5 \pm 3.1$. The standard deviation of the score is estimated by moving block bootstrap. Thus, the FCNN is outperformed by the CatBoost model which has a loss of $246.1 \pm 2.3$.

Graph Neural Networks (GNNs) are neural networks that encode the spatial dependencies in a dataset as a graph to capture spatial correlations. GNNs are natural candidates among neural networks for EV charging forecasting because they inherently encode the spatial hierarchical structure of the dataset (Wang et al., 2023; Qu et al., 2024). Among GNNs, Graph Attention Networks (GATs) are models designed for time series forecasting that exploit both temporal and spatial dependencies (Velickovic et al., 2018). Contrary to Wang et al. (2023) and Qu et al. (2024), the optimisation of this GNN did not converge, and its loss on the test set did not go below 400. We believe that this is due to the fact that we only had access to 91 charging stations, which is not a big data regime, as compared to Wang et al. (2023) who fitted their GNN on 76774 EVs and to Qu et al. (2024) who fitted their GNN on 18061 EV charging piles. Both Wang et al. (2023) and Qu et al. (2024) only had access to one month of data and focused on short-term forecasting, which may also explain this difference.

## 5 Summary of findings and discussion

This paper presents a dataset in the context of hierarchical time series forecasting of EV charging station occupancy, providing valuable insights for energy providers and EV users alike.

**Models** Contestants were able to train models that significantly outperformed the baseline performance (see Figure 8). This dataset contains many practical problems related to time series, including missing values, non-stationarity, and outliers. This explains why most contestants relied on tree-based models, which are robust enough to outperform more sophisticated machine learning methods.

**Data cleaning** Specific techniques were developed to deal with missing data and outliers (see, e.g., Section 4.1). Data preprocessing is a crucial step, and the addition of relevant exogenous features, such as the national holidays calendar, significantly improved the results.

**Time dependant loss function** All three of the winning solutions described in this paper were robust enough to maintain a high private test score, showing good generalization of the models. The choice of the empirical cost function to drive the training process produced the best results when more recent data points were given greater weight (see, e.g., Section 4.3).

**Aggregation** Aggregating the forecasts of the three winning teams even yielded a better global score, with a notable improvement at the station level. The hierarchical models presented are promising and could help improve the overall EV charging network.

**Why publishing this dataset?** This open dataset is interesting for research purpose because it encompasses many real-world problems related to time series matters, such as missing values, non-stationarities, and spatio-temporal correlations. In addition, we strongly believe that sharing the benchmark models derived from this challenge will be useful for making comparisons in future research. Two more complete datasets using new features and

spanning from July 2020 to July 2022 are available at doi.org/10.5281/zenodo.8280566 and at gitlab.com/smarter-mobility-data-challenge/additional_materials. A primary analysis is presented in the supplementary material.

**Perspectives**   Managing a fleet of EVs in the context of an increasing renewable production amount open new challenges for forecasters. We hope this dataset will allow other researchers to work on topics such as probabilistic forecasts, online learning (our challenge was "offline") or graphical models.

**Limitations**   The deployment of electric vehicles (EVs) is progressing at a remarkable pace (Sathiyan et al., 2022), making any dataset merely a snapshot of a swiftly evolving world (see also Hecht et al., 2021). To enhance forecasting accuracy, additional features could be incorporated into a dataset. Numerous covariates, such as mobility and traffic information, meteorological data, and vehicle characteristics, could be included. In a forthcoming release of the dataset, in addition to extending the observation period, we intend to incorporate traffic and meteorological data. A first attempt is proposed in the Section 4 of the supplementary material.

**Ethical concerns**   To the best of our knowledge, our work does not pose any risk of security threats or human rights violations. Knowing when and where someone plugs in their EV could lead to a risk of surveillance. However, this dataset does not contain any personal information about the user of the plug or their car, so there is no risk of consent or privacy.

## Acknowledgments

We thank Cédric Villani, Jean-Michel Poggi, and Marc Schoenauer for being part of the jury and for their insightful comments on the algorithms and on the paper. We thank Jerome Naciri, Tiphaine Phe-Neau and Jean-Yves Moise for their help in organizing the challenge. Moreover, we thank all the contestants for their original solutions. Finally, the authors thankfully acknowledge the *Manifeste IA* network of French industrial companies and the TAILOR European project on trustworthy AI for founding this challenge.

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
