# OpenReview forum: "Forecasting Electric Vehicle Charging Station Occupancy: Smarter Mobility Data Challenge"
_DMLR — Accepted by DMLR_

### Review · Reviewer_R3oD · 2024-06-12

**Recommendation:** 2
**Confidence:** 3

**Summary Of Contributions:**

In this paper, the authors introduce the charging data of electricity vehicles in Paris over seven months in 2020-2021 and evaluate the forecasts at three levels of aggregation to capture the inherent hierarchical structure of the data.

**Strengths:**

1) This paper introduces a new dataset about electric vehicles charging.
2) The authors provide detailed data processing and data analysis for the dataset.
3) The paper is easy to understand.

**Audience:**

Yes

**Broader Impact Concerns:**

No further concerns.

**Claims And Evidence:**

Yes

**Datasets And Benchmarks:**

Yes

**Extended Submissions:**

Yes

**Limitations:**

1) The main contribution of this paper is introducing a new dataset to the community. However, the novelty is still very limited as the authors only utilize very simple forecasting base model, such as ARIMA, for the prediction. The authors should implement more state-of-the-art neural networks to verify the validity and usefulness of the dataset. In addition, listing the methods of the winning teams provides almost no contribution in the academic paper. The authors should implement the state-of-the-art models and compare the results by themselves. It seems that the data is not collected by the authors but simply provided by the organization of Smarter Mobility Data Challenge. Given this, the authors should work even harder on the benchmark models and the data analysis parts.
2) Usually, the contributions of the paper are clarified at the end of the Introduction section. But the authors added a sudden contribution subsection after the Related Work.  In addition, the strengths of this paper are not convincing as compared to related works.  The authors are encouraged to reorganize the paper and improve the Related Work section.

**Requested Changes:**

1) Improve the related work.
2) Implement the state-of-the-art models to verify the validity and usefulness of the dataset.

**Strengths And Weaknesses:**

Strengths: introduce a new dataset about the electric vehicles charging.

Weaknesses:
1) Limited novelty.
2) Simple benchmark models.
3) Simply listing out the methods of the winning teams but not implementing the state-of-the-art models by the authors themselves.

---

### Review · Reviewer_CDH6 · 2024-06-20

**Recommendation:** 3
**Confidence:** 2

**Summary Of Contributions:**

This paper is a re-submission based on previous comments. I appreciate the authors for significantly revising the content and figures. It is good to see all figures are in good resolution. Overall, the paper is in good shape, and more importantly, the dataset provided will benefit future studies in this community. Therefore, I vote for acceptance of this manuscript.

**Strengths:**

[+] section 5 summarize the findings in different aspects, e.g., model and data cleaning, which benefit future studies.

[+] The topic of forecasting the occupancy of charging stations is interesting. Accurate prediction models can generate practical impacts for smart cities.

[+] The dataset and benchmark would benefit the research community in this direction.

**Audience:**

Yes

**Claims And Evidence:**

yes.

**Datasets And Benchmarks:**

The dataset provided by this paper benefit potential audience in this community.

**Extended Submissions:**

no.

**Limitations:**

[-] In Fig 8, appreciate the authors to add the model names in the x axis,  which is easy to follow. However, why two methods are missing, i.e., “tiako ianao Luna” and “MonaKaddura”.

[-] For Fig 1, the red line in the left figure (stations and number of observations) looks weird.

[-] In Section 3, Problem description, the authors did a great job in introducing the detailed setting. However, at the beginning of this section, it will further enhance the readability if one paragraph can be added to introduce the goal of the challenge. For example, “In this section, we introduce the hierarchical forecasting challenge proposed to the contestants
of the Smarter Mobility Challenge. The overall goal is xxx. Accurate prediction of xxx benefits xxx. ”

**Requested Changes:**

please refer to the limitations.

**Strengths And Weaknesses:**

please see detailed comments below.

---

### Review · Reviewer_nNmw · 2024-07-24

**Recommendation:** 4
**Confidence:** 2

**Summary Of Contributions:**

1. An open dataset on electric vehicle behaviors gathering both spatial and hierarchical features. Datasets with such features are rare and valuable for electric network management.
2. An in-depth descriptive analysis of this dataset revealing meaningful user behaviors .
3. A detailed and reproducible benchmark for forecasting the EV charging station occupancy. This benchmark compares the winning solutions of a data challenge and state-of-the-art predictive models.

**Strengths:**

1. This dataset was adopted in Smarter Mobility Data Challenge, the solutions submitted by winning teams were also described in the paper, which can be used as baselines for addressing similar tasks.
2. Additionally, since users' EV charging strategy may interact with pricing scheme of stations, the authors also provide detailed descriptions about the pricing scheme.

**Audience:**

Yes

**Claims And Evidence:**

The claims in the submission are supported by accurate, convincing and clear evidence.

**Datasets And Benchmarks:**

There is sufficient detail on data collection and organization, availability and maintenance, and ethical and responsible use. The methods are described in detail.

**Extended Submissions:**

Not applicable

**Limitations:**

From operational perspective, the ability of uncertainty quantification is of great importance and should be an important aspect in benchmarking.

**Requested Changes:**

All my questions and concerns have been properly addressed in this submission.

**Strengths And Weaknesses:**

Strengths:
1. Clarity of description of data collection and partition is improved In the latest version of manuscript, the dataset, baseline models as well as aggregations are described in details, and a gitlab link is also provided.
2. The winning solutions are properly described and a range of benchmarking methods are considered, including SARIMAX, LSTM, XGBoost, random forest, and CatBoost.
3. The dataset has a natural hierarchical structure, i.e., individual stations, areas and global, and is relevant to many research areas.

Weaknesses: From operational perspective, the ability of uncertainty quantification is of great importance and should be an important aspect in benchmarking.